# Level of overweight and obesity surpassed underweight among women in 40 low and middle-income countries: Findings from a multilevel multinomial analysis of population survey data

**Kusse Urmale Mare** [1]*, **Kebede Gemeda Sabo**[1], **Beriso Furo Wengoro**[2], **Begetayinoral Kussia Lahole**[3]

**1** Department of Nursing, College of Medicine and Health Sciences, Samara University, Samara, Ethiopia, **2** Department of Biomedical Sciences, College of Medicine and Health Sciences, Samara University, Samara, Ethiopia, **3** Department of Midwifery, College of Medicine and Health Sciences, Arba Minch University, Arba Minch, Ethiopia

* kussesinbo@gmail.com

## Abstract

### Background

Despite continued global and local initiatives to address nutritional problems, low- and middle-income countries are facing both malnutrition and non-communicable diseases, with about 80% of non-communicable disease-related deaths. There is a dearth of recent evidence on the extent and determinants of underweight, overweight, and obesity in this region, which is essential for guiding intervention programs. Thus, this study intended to provide insights into the current level of malnutrition among women of reproductive age in low- and middle-income countries.

### Methods

A secondary analysis of Demographic and Health Survey data from 40 low- and middle-income countries was performed using a weighted sample of 1,044,340 women of reproductive age. Forest plots and line graph were used to present the pooled and regional estimates of underweight and overweight and obesity. A multilevel multinomial logistic regression models were fitted to identify determinants of malnutrition and models were compared based on the deviance and log-likelihood values. In the final model, statistical significance was determined using an adjusted relative risk ratio with the corresponding 95% confidence interval.

### Results

The overall prevalence of obesity and overweight was 26.8% (95% CI: 26.7%–26.9%), with the highest rate in Jordan (67.2%) and lowest in Burundi (9.5%). Additionally, the level of underweight was found to be 15.6% [95% CI: 15.5%–15.7%], ranging from 1.3% in

**Data availability statement:** The data used and analyzed in this study can be accessed from the DHS website (https://dhsprogram.com/data/dataset_admin/index.cfm).

**Funding:** The author(s) received no specific funding for this work.

**Competing interests:** The authors have declared that no competing interests exist.

**Abbreviations:** ARRR, Adjusted Relative Risk Ratio; AIC BMI, Body Mass Index; DHS, Demographic and Health Survey; GVIF, Generalized Variance Inflation Factor; ICC, Intra-class correlation; LL, Log-likelihood; LMICs, Lower- and Middle-Income Countries; MOR, Median Odds Ratio; NCD, Non-communicable Diseases; PCV, Proportion Change in Variance; SSA, Sub-Saharan Africa; WHO, World Health Organization; WRA, Women of Reproductive Age.

Jordan to 25.4% in Timor-Leste. Women of families with middle and rich wealth indexes, those aged 25–34 and 35–49, contraceptive users, those with primary and higher education, and urban residents were more likely to be overweight or obese. In contrast, the results showed a lower likelihood of undernutrition among women in the households with middle [ARRR (95% CI): 0.83 (0.80–0.86)] and rich wealth indexes [ARRR (95% CI): 0.73 (0.71–0.76)], those with primary [ARRR (95% CI): 0.70 (0.68–0.73)], secondary [ARRR (95% CI): 0.71 (0.69–0.74)], and higher education [ARRR (95% CI): 0.57 (0.53–0.61))], and women with media access [ARRR (95% CI): 0.79 (0.77–0.82)].

## Conclusion

Over a quarter of women of reproductive age in LMICs were overweight or obese, with underweight affecting 15.6%. It was also found that the levels of overweight and obesity were higher than the rate for underweight, with wide variations across the countries. Thus, efforts to improve the modifiable risks can have a positive impact on reducing undernutrition and over-nutrition. Given the considerable variations between countries and survey periods, nutrition programs should also be tailored to the specific needs and cultural contexts of different regions within the study settings. Furthermore, the findings suggest the need for regular monitoring and evaluation of the existing nutritional interventions to assess their effectiveness.

## Background

Overweight and obesity are significant public health challenges in both developed and developing regions, disproportionately affecting women (40%) than men (35%) [1,2]. While overweight is marked by an excessive accumulation of body fat, obesity is a chronic and complex condition with substantial fat deposition. Both conditions pose varying degrees of health risks associated with excessive body fat [1]. There are several linkages between obesity and health problems, including type 2 diabetes and heart disease, poor bone health, disturbed sleep, impaired reproductive function and mobility, and cancers of the reproductive organs [1,3,4].

Excessive weight gain in women is also associated with infertility, exacerbated polycystic ovary syndrome, heart disease, diabetes, and breast cancer due to changes in reproductive hormones [5,6]. Being overweight and obese during pregnancy is also linked with a higher risk of gestational diabetes and hypertension [7,8], and complications like postpartum hemorrhage, assisted delivery, and surgical site infections [9–11]. Furthermore, maternal exposure to these nutritional indices during the prenatal period elevates the risk of congenital malformations, preterm birth, perinatal death, and macrosomia [5,12,13].

Globally, about 4.0 million deaths and 120 million disability-adjusted life-years are attributed to overweight and obesity [14]. In 2022, approximately 1 in 8 individuals worldwide were living with obesity and 2.5 billion adults aged 18 years and older were overweight, with 890 million affected by obesity [1]. Additionally, 43% of women worldwide are affected by overweight or obesity [1].

The prevalence of overweight and obesity is on the rise in developing countries alongside economic development, reduced physical activity, and a shift towards high-energy and fatty diets [15,16]. For instance, among women, the level of overweight varies from 10% to 60% [17–25], while the prevalence of obesity ranges from 3% to 35% [17,20,23,26].

Previous studies showed that the factors associated with overweight and obesity include gender [8,27,28], age [8,27,29,30], marital status [26,27,29,32], wealth status [8,28,31],

exposure to media [33,8], physical activities [28,8], alcohol consumption [31,8], snacking [31], education [29,30], occupation [30], contraceptive use [30], and place of residence [29,32].

Despite a continued effort of the International Fund for Agricultural Development and World Health Organization (WHO) to address the increasing rates of obesity and overweight, particularly in low- and middle-income countries (LMICs) [1,34], these regions are facing a double burden of malnutrition, where approximately 70% of the population are overweight or obese [34]. Additionally, these settings face an increased risk of non-communicable diseases (NCDs) such as diabetes and cardiovascular diseases, with about 80% of NCD-related deaths [34]. Despite these alarming figures, there is a lack of up-to-date evidence on the level and determinants of overweight and obesity in this region. Furthermore, previous studies in the region have utilized outdated DHS data, whereas our study aimed to bridge this gap by incorporating the most recent DHS datasets from the countries included. Thus, this study intended to provide insights into the current level of overweight and obesity among women of reproductive age (WRA) in LMICs using population survey data from 40 countries.

## Methods

### Data source and participants

Data were obtained from demographic and health surveys conducted between 2015 and 2022 in 40 LMICs (https://dhsprogram.com/data/dataset-admin/index.cfm). Before inclusion, the survey dataset of each country was checked for the availability of body mass index (BMI) records, and countries with no data on this record were excluded from the analysis. After dropping records with missing observations and those with no unknown BMI data, a weighted sub-sample of 1,044,340 WRA (unweighted sample = 1,052,841) in all surveys was included in the analysis.

### Variables and measurement

**Dependent variable.** The dependent variable was BMI, measured as the ratio of the weight in kilograms to the square of the height in meters ($kg/m^2$). Based on the WHO cut-off point, BMI was classified as underweight, normal, overweight, and obese if the BMI is $< 18.5\,kg/m^2$, $18.5–24.9\,kg/m^2$, $25–29.9\,kg/m^2$, and $\geq 30\,kg/m^2$, respectively [35]. For the analysis purpose, overweight and obesity were treated as one category.

**Independent variables.** Individual-level (level-1) variables included age, marital status, household wealth, health insurance coverage, women's and partner's education, employment, marital status, media exposure, household head, age at marriage, history of pregnancy loss, parity, number of under-five children, birth order, preceding birth interval, anemia status, contraceptive use, decision maker on women's health, source of drinking water, and toilet facility. While, region, place of residence, and distance to health facility were the community-level (level-2) variables.

Media exposure was constructed using the frequency of watching television, listening to the radio, and reading newspapers, which have three response options (i.e., not at all, less than once a week, and at least once a week). Using the DHS definition, women who indicated that they watched television, listened to the radio, or read the newspaper at least once a week were classified as having media exposure, while those who did not meet this criterion were classified as not having exposure.

### Data management and analysis

Data cleaning and analysis were conducted using Stata software version 17. Before analysis, the presence of the outcome variable (i.e., BMI) in the DHS dataset of each country was

confirmed, and all study variables were assessed for missing data. Records with missing observations and unknown BMI were dropped from the analysis. Finally, the datasets of 40 LMICs were appended and weighted to compensate for the non-representativeness of the sample and obtain reliable estimates and standard errors.

To account for the hierarchical structure of DHS design and the polytomous nature of the outcome variable, a multilevel multinomial logistic regression model was applied to determine the effects of independent variables on underweight and overweight and obesity. Initially, bivariable multilevel logistic regression analysis was done and variables with a p-value of less than 0.25 in this analysis were considered for the multivariable regression model.

## Fixed-effect multilevel analysis

Four hierarchical models were independently fitted to select the model that best fits the data. To test the random variability in the intercept, an empty model, i.e., a model without an independent variable was fitted. In the second and third models, the effect of individual and community level variables on abnormal BMI was respectively assessed, while the combined effect of both community and individual level variables on the outcome variable was simultaneously assessed in the fourth model.

In Model I (empty model), the likelihood of being underweight and overweight and obese is only a function of the clusters in which the women lived, which is indicated by a random intercept at the cluster level. In Model II (a model with level-1 variables), the probability of being underweight and overweight and obese was measured as a function of the cluster in which the women resided and individual-level characteristics. Model III (model with level-2 variables) assessed the probability of malnutrition as a function of cluster and community-level variables, while Model IV (model with both level-1 and level-2 variables) examined the combined effect of both level-1 and level-2 variables on the occurrence of underweight, overweight and obesity.

After fitting these models, models were compared based on log-likelihood (LL) and deviance (i.e., -2*LL) values to select the one that best fits the data. For instance, model IV has the lowest deviance and highest LL values and is hence considered the best-fit model for the final analysis. A generalized variance inflation factor [GVIF = VIF$^{[1/(2*df)]}$] was used to determine the presence of collinearity between explanatory variables and this value was less than five for all variables included in the final model, suggesting the non-existence of multi-collinearity. Finally, in the multivariable analysis, a p-value less than 0.05 and an adjusted relative risk ratio (ARRR) with the corresponding 95% confidence interval were used to identify determinants of underweight, overweight, and obesity.

## Random-effect analysis (measures of variance)

Random variability in the level of underweight, overweight, and obesity across clusters was assessed by intra-class correlation coefficient (ICC), proportion change in variance (PCV), and median odds ratio (MOR). In this analysis, ICC indicates the total variance in the proportion of underweight, overweight, and obesity attributed to cluster differences. PCV (attributable percentage variability in the multilevel model) measured the differences in the prevalence of malnutrition attributed to community or individual-level characteristics. Heterogeneity in the risk of underweight and overweight and obesity among women in different clusters was assessed by MOR. It was measured as the median value of the odds of the outcome variable (i.e., abnormal BMI) between women living in the area at the highest risk of abnormal BMI compared to those in the clusters at the lowest risk of the event.

### Ethical approval

Since we used secondary data, permission to access the data was granted from the Measure DHS official Database via an online request at http://www.dhsprogram.com. The survey procedures were also approved by the Institutional Review Board of the host country and ICF International. Additionally, written informed consent was initially obtained from the study participants during the collection of the survey data.

## Results

### Sociodemographic characteristics

Of 1,044,340 women of reproductive age, about three-fourths (75.2%) were from the Asia region, more than half (56.0%) resided in rural areas, and 433,278 (41.5%) were from rich households. Moreover, 357,587 (34.2) were between the ages of 15-24 years, less than one-quarter (22.8%) had no formal schooling, and 733,430 (70.2%) were currently in a union (Table 1).

In this analysis, the highest prevalence of underweight was observed in women from Asia (17.0%), rural settings (17.4%), poor households (19.9%), and younger age groups (35.0%). Conversely, women from the African region (27.7%), urban dwelling (36.7%), rich households (35.7%), older age groups (37.6%), and those with higher education (31.0%) had higher rates of overweight and obesity than their respective categories (Table 1).

### Reproductive and health-related characteristics

More than half (58.3%) of women were married at 18 years of age or older, 136,858 (13.1%) had experienced pregnancy loss, and 479,261 (45.9%) were multiparous. Additionally, about half (51.9%) of WRA were anemic, 459,607 (44.0%) were using contraceptive methods, and nearly a quarter (26.7%) reported that distance to the nearest health facility was a big problem.

The highest proportion of undernutrition was observed among nulliparous women (24.7%), women with a birth order of 4 or more (19.8%), anemic women (18.1%), and those not using contraception (18.4%). Conversely, multiparous women (32.4%), those with a birth interval of at least 33 months (34.4%), and women using contraceptives (31.2%) had comparatively higher levels of overweight and obesity. Furthermore, the prevalence of overweight and obesity was higher among women with a history of pregnancy loss (37.2%), those with a lower birth order (31.1%), and those who made their own healthcare decisions (40%) (Table 2).

### Prevalence of underweight

Our result showed that the overall prevalence of underweight among WRA in LMICs was 15.6% [95% CI: 15.5%–15.7%], with large variations across the included countries. The highest level of undernutrition was observed in Timor-Leste (25.4%), Ethiopia (21.1%), Chad (17.9%), and Burundi (17.6%), while Jordan (1.3%), Guatemala (2.9%), and Zambia (3.0%) had the lowest rates. Moreover, subgroup analysis revealed that Asia had the highest prevalence of underweight (17.0%), followed by Africa (11.2%) (Fig 1).

### Prevalence of overweight and obesity

The pooled prevalence of obesity and overweight among WRA in LMICs was 26.8% (95% IC: 26.7%–26.9%), of which overweight and obesity accounted for 18.8% [95% CI: 18.7%–18.9%] and 8.0% [95% CI: 7.9%–8.1], respectively. Higher rates were found in Jordan (67.2%),

**Table 1. Socio-economic characteristics of women of reproductive age in 40 low and middle-income countries, 2015 to 2022.**

| Characteristics | Weighted frequency (%) | Underweight (95% CI) | Overweight and obesity (95% CI) |
|---|---|---|---|
| Age | | | |
| 15–25 year | 357,587 (34.2) | 25.0 [24.9–25.2] | 11.0 [10.9–11.1] |
| 25–34 year | 319,714 (30.6) | 12.0 [11.9–12.1] | 27.7 [27.5–27.8] |
| >=35 years | 367,038 (35.2) | 8.9 [8.8–9.0] | 37.6 [37.4–37.7] |
| Educational status | | | |
| No formal education | 237,726 (22.8) | 15.9 [15.8–16.1] | 21.3 [21.2–21.5] |
| Primary education | 185,877 (17.8) | 12.4 [12.2–12.5] | 28.7 [28.5–28.9] |
| Secondary education | 480,398 (46.0) | 17.1 [17.0–17.2] | 24.9 [24.7–25.0] |
| Higher education | 140,336 (13.4) | 11.9 [11.7–12.1] | 31.0 [30.7–31.3] |
| Working status (n = 455,170) | | | |
| Non–working | 252,205 (55.4) | 13.9 [13.7–14.0] | 28.4 [28.3–28.6] |
| Working | 202,965 (44.6) | 10.0 [9.9–10.1] | 31.8 [31.6–32.0] |
| Marital status | | | |
| Never in union | 254,214 (24.3) | 26.3 [26.1–26.5] | 11.3 [11.2–11.4] |
| Currently in union | 733,430 (70.2) | 11.7 [11.6–11.8] | 30.0 [29.9–30.1] |
| Formerly in union | 56,695 (5.5) | 11.1 [10.9–11.4] | 32.6 [32.2–33.0] |
| Media exposure | | | |
| No | 446,154 (42.7) | 18.1 [18.0–18.2] | 18.5 [18.4–18.7] |
| Yes | 598,186 (57.3) | 13.1 [13.0–13.2] | 31.1 [31.0–31.2] |
| Covered by health insurance (n = 993,367) | | | |
| No | 738,086 (74.3) | 15.1 [15.0–15.2] | 26.6 [25.5–25.7] |
| Yes | 255,281 (25.7) | 15.9 [15.7–16.0] | 25.3 [25.1–25.5] |
| Household head | | | |
| Male | 839,056 (80.3) | 15.6 [15.5–15.6] | 25.0 [24.9–25.1] |
| Female | 205,284 (19.7) | 14.5 [14.3–14.6] | 27.3 [27.0–27.4] |
| Household wealth | | | |
| Poor | 397,860 (38.1) | 19.9 [19.7–20.0] | 16.4 [16.2–16.5] |
| Middle | 213,202 (20.4) | 14.8 [14.6–14.9] | 25.3 [25.1–25.5] |
| Rich | 433,278 (41.5) | 10.6 [10.5–10.7] | 35.7 [35.6–35.9] |
| Place of residence | | | |
| Urban | 365,235 (35.0) | 10.5 [10.4–10.6] | 36.7 [36.5–36.8] |
| Rural | 679,104 (56.0) | 17.4 [17.3–17.5] | 20.7 [20.6–20.8] |
| Region | | | |
| Africa | 203,839 (19.5) | 11.2 [11.1–11.3] | 27.7 [27.5–27.9] |
| Asia | 796,119 (75.2) | 17.0 [16.9–17.1] | 23.8 [23.7–23.9] |
| Others* | 44,381 (4.2) | 4.8 [4.6–5.0] | 4.6 [4.5–4.6] |

*Others: Latin America, Caribbean, and Europe.

Turkey (62.2%), and Zambia (61.6%), while the lowest prevalence was noted in Burundi (9.5%) and Timor-Leste (9.9%). Additionally, the analysis noted the highest prevalence of overnutrition among women in Latin America, Caribbean, and Europe (42.7%), followed by Africa (40.6%) (Fig 2).

Of forty countries included in the analysis, the rate of overweight and obesity exceeded the prevalence of underweight in 35 countries. The exceptions were five countries (Burundi, Ethiopia, Madagascar, Chad, and Timor-Leste) where the highest level of undernutrition was observed (Table 3).

**Table 2. Reproductive and health-related characteristics of women of reproductive age in 40 low and middle-income countries, 2015 to 2022.**

| Characteristics | Weighted frequency (%) | Underweight (95% CI) | Overweight and obesity (95% CI) |
|---|---|---|---|
| Age at marriage (n = 790,125) | | | |
| < 18 year | 329,519 (41.7) | 13.2 [13.1–13.4] | 27.3 [27.2–27.5] |
| >= 18 year | 460,606 (58.3) | 10.6 [10.6–10.7] | 32.1 [32.0–32.2] |
| History of pregnancy loss | | | |
| No | 907,456 (86.9) | 16.1 [16.0–16.2] | 23.8 [23.7–23.9] |
| Yes | 136,858 (13.1) | 10.1 [9.9–10.3] | 37.2 [36.9–37.4] |
| Parity | | | |
| Nullipara | 310,076 (29.7) | 24.7 [24.5–24.8] | 12.7 [12.6–12.9] |
| Primipara | 151,396 (14.5) | 12.7 [12.5–12.8] | 26.9 [26.6–27.1] |
| Multipara | 479,261 (45.9) | 10.8 [10.7–10.9] | 32.4 [32.2–32.6] |
| Grand-multipara | 103,607 (9.9) | 11.8 [11.7–12.0] | 30.2 [30.0–30.5] |
| Number of U5 children | | | |
| 0 | 587,244 (56.2) | 15.8 [15.7–15.9] | 26.3 [26.2–26.4] |
| 1 | 266,948 (25.6) | 14.2 [14.1–14.3] | 26.2 [26.0–26.3] |
| 2+ | 190,148 (18.2) | 15.6 [15.4–15.7] | 21.8 [21.6–22.0] |
| Birth order | | | |
| 1 to 3 | 548,478 (52.5) | 11.2 [11.1–11.3] | 31.1 [30.9–31.2] |
| >=4 | 495,862 (47.4) | 19.8 [19.6–19.9] | 19.6 [19.5–19.7] |
| Preceding birth interval (n = 580,547) | | | |
| < 33 months | 274,304 (47.2) | 12.1 [11.9–12.2] | 29.3 [29.1–29.5] |
| >= 33 months | 306,243 (52.8) | 10.0 [9.9–10.1] | 34.4 [34.3–34.6] |
| Anemia status (n = 951,070) | | | |
| No anemic | 457,754 (48.1) | 12.9 [12.8–13.0] | 28.3 [28.2–28.5] |
| Anemia | 493,316 (51.9) | 18.1 [17.9–18.2] | 21.7 [21.6–21.8] |
| Contraceptive use | | | |
| Not using | 584,733 (56.0) | 18.4 [18.3–18.5] | 21.2 [21.1–21.3] |
| Using | 459,607 (44.0) | 11.2 [11.1–11.3] | 31.2 [31.0–31.3] |
| Perceived distance to health facility (1,026,269) | | | |
| Not a big problem | 752,281 (73.3) | 14.8 [14.7–14.9] | 27.0 [26.9–27.1] |
| Big problem | 273,988 (26.7) | 16.8 [16.6–16.9] | 27.1[21.4–21.7] |
| Decision maker on women's health | | | |
| Woman alone | 57,115 (5.5) | 7.8 [7.6–8.0] | 40.0 [39.6–40.4] |
| Jointly with partner | 164,450 (15.7) | 9.4 [9.3–9.6] | 36.1 [35.8–36.3] |
| Others* | 822,774 (78.8) | 17.1 [17.0–17.1] | 22.3 [22.3–22.4] |
| Toilet facility | | | |
| Unimproved | 290,319 (27.8) | 19.8 [19.6–19.9] | 16.8 [16.7–16.9] |
| Improved | 754,021 (72.2) | 13.7 [13.6–13.7] | 28.8 [28.7–28.9] |
| Drinking water source | | | |
| Unimproved | 145,724 (14.0) | 15.4 [15.2–15.5] | 21.4 [21.2–21.6] |
| Improved | 898,615 (86.0) | 15.4 [15.3–15.4] | 26.2 [26.1–26.3] |

*Others: husband only; someone else; other DHS category.

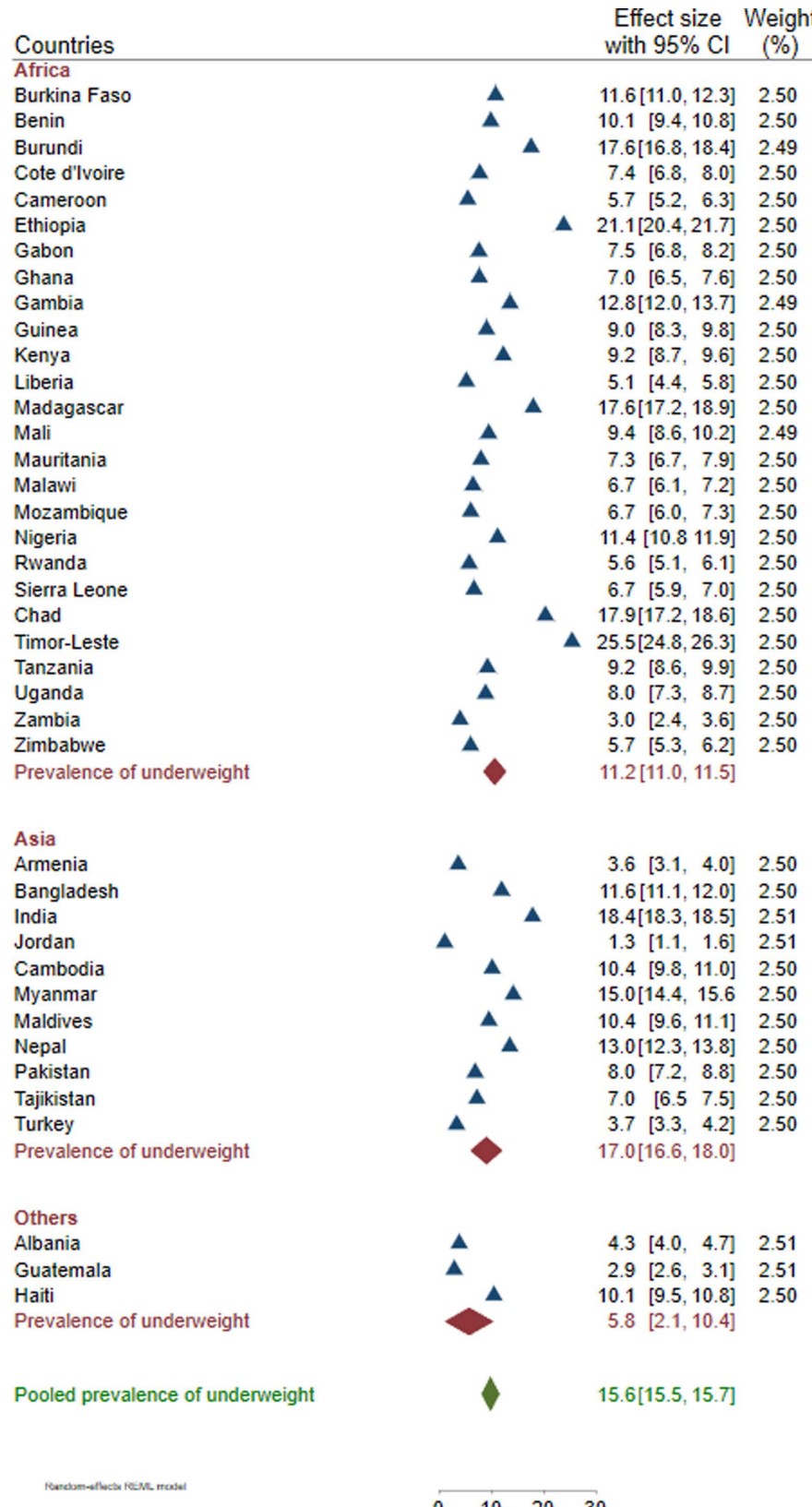

**Fig 1. Pooled and national estimates of underweight among women of reproductive age in 40 low- and middle-income countries from 2015 to 2022.**

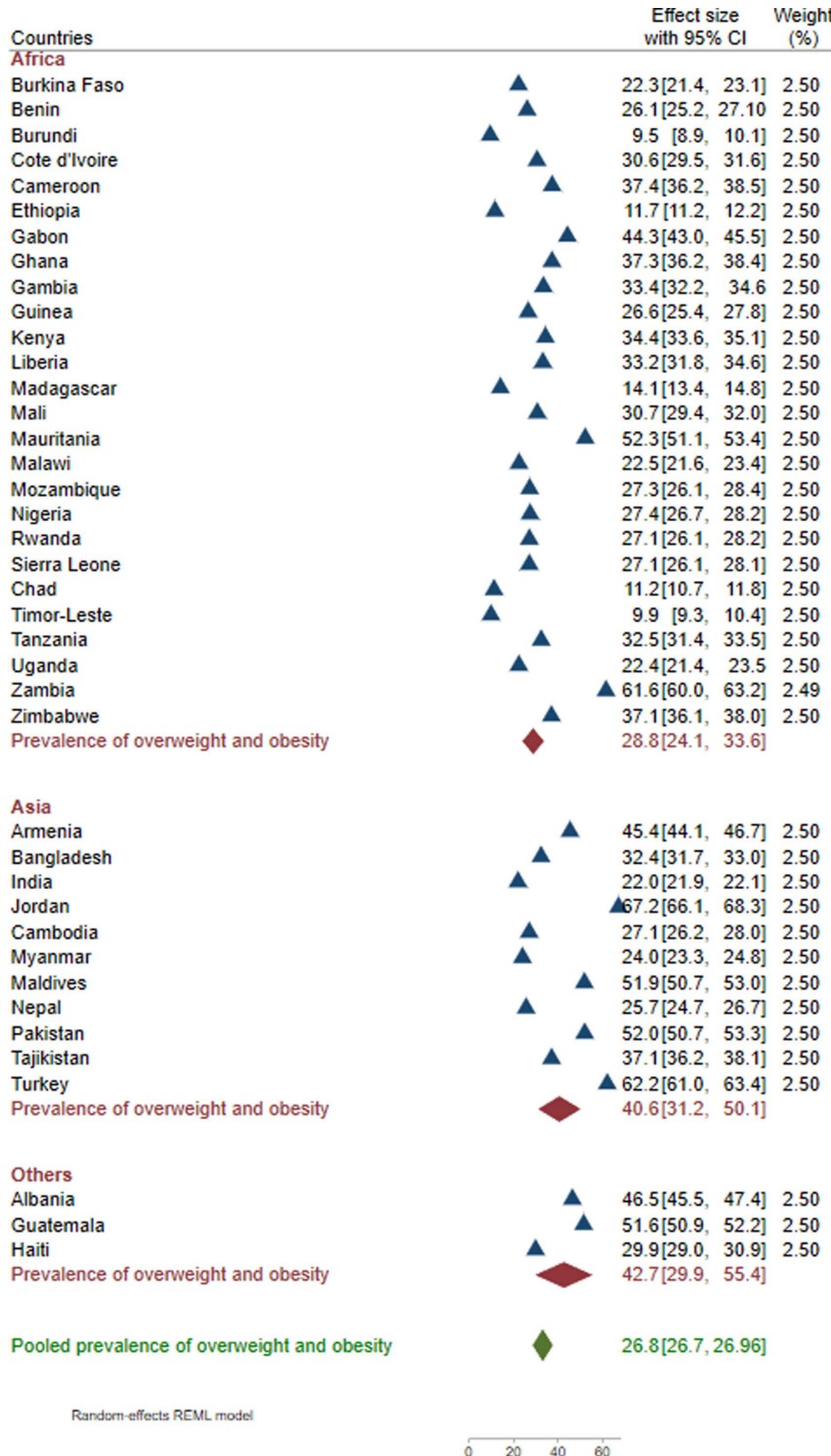

**Fig 2. Pooled and national estimates of overweight and obesity among women of reproductive age in 40 low- and middle-income countries from 2015 to 2022.**

**Table 3. Country-specific prevalence of underweight and overweight and obesity among women of reproductive age in 40 low- and middle-income countries, 2015 to 2022.**

| Country | Survey year | Underweight | Overweight and obesity | Total |
|---|---|---|---|---|
| Albania | 2018 | 463 (4.3) | 4863 (45.5) | 10668 |
| Armenia | 2016 | 210 (3.6) | 2638 (44.9) | 5877 |
| Bangladesh | 2018 | 2285 (11.5) | 6328 (32.0) | 19786 |
| Burkina Faso | 2021 | 1030 (11.6) | 1923 (21.7) | 8852 |
| Benin | 2018 | 815 (10.1) | 2094 (26.0) | 8065 |
| Burundi | 2017 | 1520 (17.6) | 728 (8.5) | 8620 |
| Cote d'Ivoire | 2021 | 533 (7.4) | 2400 (33.2) | 7214 |
| Cameroon | 2018 | 395 (5.7) | 2575 (37.3) | 6892 |
| Ethiopia | 2016 | 3188 (21.1) | 1184 (7.8) | 15123 |
| Gabon | 2019 | 458 (7.5) | 2968 (48.6) | 6111 |
| Ghana | 2022 | 544 (7.0) | 3364 (43.5) | 7724 |
| Gambia | 2019 | 762 (12.8) | 2154 (36.2) | 5952 |
| Guinea | 2021 | 480 (9.0) | 1404 (26.3) | 5322 |
| Guatemala | 2015 | 688 (2.9) | 12533 (51.8) | 24184 |
| Haiti | 2016 | 966 (10.1) | 3074 (32.3) | 9530 |
| India | 2021 | 127250 (18.4) | 165307 (23.9) | 692238 |
| Jordan | 2017 | 94 (1.3) | 4911 (68.6) | 7155 |
| Kenya | 2022 | 1513 (9.2) | 6317 (38.3) | 16486 |
| Cambodia | 2022 | 1024 (10.4) | 2811 (28.6) | 9833 |
| Liberia | 2019 | 208 (5.1) | 1513 (37.1) | 4083 |
| Madagascar | 2021 | 1680 (17.6) | 1339 (14.0) | 9553 |
| Mali | 2018 | 480 (9.4) | 1416 (27.6) | 5119 |
| Myanmar | 2015 | 1898 (15.0) | 3133 (24.8) | 12640 |
| Mauritania | 2019 | 538 (7.3) | 3942 (53.5) | 7363 |
| Maldives | 2016 | 724 (10.3) | 3485 (49.8) | 6998 |
| Malawi | 2015 | 534 (6.7) | 1713 (21.4) | 8009 |
| Mozambique | 2022 | 398 (6.7) | 1405 (23.6) | 5955 |
| Nigeria | 2018 | 1672 (11.3) | 4145 (28.2) | 14730 |
| Nepal | 2022 | 961 (13.0) | 2154 (29.2) | 7370 |
| Pakistan | 2017 | 350 (8.0) | 2254 (51.2) | 4400 |
| Rwanda | 2019 | 405 (5.6) | 1991 (27.5) | 7265 |
| Sierra Leone | 2019 | 486 (6.5) | 2116 (28.1) | 7530 |
| Chad | 2015 | 2029 (17.9) | 1348 (11.9) | 11355 |
| Tajikistan | 2017 | 744 (7.0) | 3983 (37.3) | 10669 |
| Timor-Leste | 2016 | 3180 (25.5) | 1280 (10.2) | 12451 |
| Turkey | 2018 | 250 (3.7) | 4003 (59.7) | 6701 |
| Tanzania | 2022 | 696 (9.2) | 2338 (30.9) | 7570 |
| Uganda | 2016 | 481 (8.0) | 1464 (24.3) | 6012 |
| Zambia | 2018 | 97 (3.0) | 2025 (62.3) | 3251 |
| Zimbabwe | 2015 | 554 (5.7) | 3446 (35.5) | 9684 |

## Change in the level of underweight and overweight and obesity over survey years

Fig 3 shows the change in the level of underweight and overweight and obesity across the survey years in 40 LMICs. This result indicated that the level of obesity and overweight surpassed

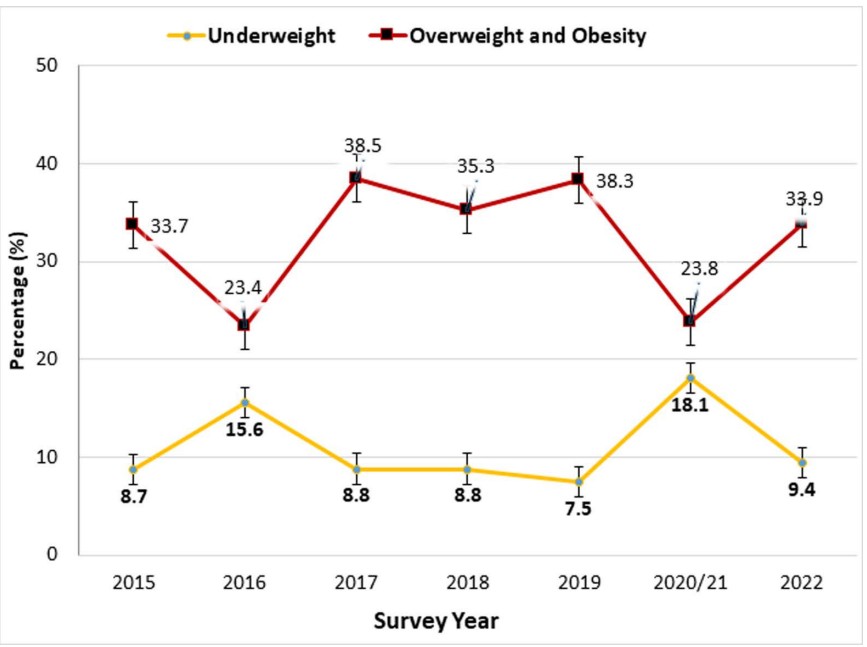

**Fig 3. Change in the level of underweight and overweight and obesity among women of reproductive age in 40 low- and middle-income countries from 2015 to 2022.**

that of underweight during all survey periods, with the peak prevalence in 2017 (38.5%) and 2019 (38.3%). Our analysis also showed a decline in undernutrition from 18.1% to 9.4% between 2020/21 and 2022, while the level of overweight and obesity rose by about 10%, from 23.8% to 33.9% during the same period. Overall, both undernutrition and overnutrition were slightly increased between 2015 and 2022 (Fig 3).

## Random effect result (measures of variations)

A random effect analysis result indicated that 44% of the variability in the level of malnutrition was due to differences at the cluster level (as indicated by the ICC value of model 1), while 31% of the total variability was explained by both individual and community-level factors (ICC value of model IV). In the final model (model IV), the PCV value also shows that 29% of the variation in malnutrition among WRA was explained by the effect of both individual and community-level covariates. Furthermore, the MOR of an empty model indicated the existence of heterogeneity in the magnitude of underweight and overweight and obesity across clusters. This implies that WRA in the clusters with higher levels of malnutrition were 1.72 times more likely to suffer from malnutrition than those in the clusters with lower prevalence (Table 4).

## Determinants of underweight and overweight and obesity

The result showed that the risk of overweight and obesity was higher for WRA of households with middle [ARRR (95% CI): 1.24 (1.22–1.27)] and rich wealth indexes [ARRR (95% CI): 1.49 (1.46–1.52)] and those aged 25–34 [ARRR (95% CI): 2.08 (2.02–2.13)] and 35–49 years [ARRR (95% CI): 3.56 (3.35–3.66)]. Moreover, contraceptive use [ARRR (95% CI): 1.22 (1.20–1.25)], attending primary [ARRR (95% CI): 1.45 (1.42–1.48)], secondary [ARRR (95% CI): 1.68 (1.64–1.72)] and higher education [ARRR (95% CI): 1.75 (1.69–1.81)], and living

**Table 4. Measures of variations and model statistics summary for determinants of underweight and overweight and obesity among women of reproductive age in 40 LMICs, 2015 to 2022.**

| Measure of variation | Model 1 | Model 2 | Model 3 | Model 4 |
|---|---|---|---|---|
| Cluster-level variance | 0.44 | 0.27 | 0.23 | 31 |
| Intra-class correlation | 0.26 | 0.28 | 0.17 | 0.22 |
| Explained variance (PCV) | Reference | 0.39 | 0.47 | 0.29 |
| Median odds ratio | 1.72 | 1.35 | 1.24 | 1.44 |
| **Model statistics summary** | | | | |
| Log-likelihood | −242,243 | −156,670 | −242,128 | −156,148 |
| Deviance | 484,486 | 313,340 | 482,256 | 312,296 |

in urban areas [ARRR (95% CI): 1.66 (1.63–1.69)] were associated with an increased risk overnutrition.

Our result also revealed that the likelihood of undernutrition was lower for women in the households with middle [ARRR (95% CI): 0.83 (0.80–0.86)] and rich wealth indexes [ARRR (95% CI): 0.73 (0.71–0.76)], those with primary [ARRR (95% CI): 0.70 (0.68–0.73)], secondary [ARRR (95% CI): 0.71 (0.69–0.74)] and higher education [ARRR (95% CI): 0.57 (0.53–0.61)], and women with media access [ARRR (95% CI): 0.79 (0.77–0.82)]. Additionally, age, employment status, media exposure, women's involvement in healthcare decisions, parity, contraceptive use, and place of residence were significantly associated with undernutrition (Table 5).

## Discussion

This study assessed the level and determinants of underweight, overweight, and obesity among WRA in 40 LMICs. The pooled prevalence of overweight and obesity was 26.8% (95% CI: 26.7%–26.9%). The current result is in line with prior studies in England (26%) [36] and Mali (26.9%) [37], but higher than the prevalence reported from the studies done in Ethiopia (15%) [24], Malawi (23%) [38], Sub-Saharan Africa (22.6%) [39,40], and Uganda (22.4%) [18]. Contrarily, our finding is lower compared to the studies conducted in South Africa (27%) [41], Nigeria (30%) [42], Cameroon (50%) [43], Egypt (85%) [44], Kenya (30%) [30], Zimbabwe (46.5%) [45], and Australia (60%) [46]. These disparities could be attributed to divergent levels of economic development, urbanization promoting sedentary lifestyles, culture-specific food taboos and dietary practices, and variations in levels of physical activity and nutritional awareness across the study settings.

Our findings also revealed that the prevalence of underweight among WRA in LMICs was 15.6% [95% CI: 15.5%–15.7%], with substantial variations among the countries studied. This finding is lower than the rates reported in India (20%) [47], South Asia (24%) [48], and LMICs (42%) [48]. However, our finding is higher than that reported in the studies done in Bangladesh (12%) [49], Sierra Leone (6.7%) [50], global prevalence (9.7%) [51], Myanmar (14%) [52], Indonesia (14%) [53], and Botswana (10%) [54]. These variations can be attributed to differences in socioeconomic status, disparities in education and health awareness, cultural dietary practices, regional food security discrepancies, and access to nutritional education across different settings.

Similar to the results of the studies done in Kenya [55], Bangladesh [56], India [57], Nepal [56], and SSA [58], our analysis revealed that women from households with middle and rich wealth indexes exhibited a higher risk of overweight and obesity compared to their counterparts. Our study also found that belonging to wealthy families is associated with lower odds of being underweight compared to women of families with poor wealth index. This aligns with previous studies in various contexts [59–61] showing similar associations. This could be

**Table 5. Determinants of underweight and overweight and obesity among women of reproductive age in 40 low and middle-income countries, 2015 to 2022.**

| Characteristics | Underweight ARRR (95% CI) | Overweight and Obesity ARRR (95% CI) |
|---|---|---|
| Age at marriage | | |
| < 18 year | Ref | Ref |
| >= 18 year | 0.94 (0.93, 0.97)* | 0.94 (0.93, 0.96)** |
| Age | | |
| 15–25 year | Ref | Ref |
| 25–34 year | 0.80 (0.78, 0.83)* | 2.08 (2.02, 2.13)* |
| >=35 years | 0.74 (0.71, 0.76)* | 3.56 (3.35, 3.66)* |
| Educational status | | |
| No formal education | Ref | Ref |
| Primary education | 0.70 (0.68, 0.73)* | 1.45 (1.42, 1.48)** |
| Secondary education | 0.71 (0.69, 0.74)* | 1.68 (1.64, 1.72)** |
| Higher education | 0.57 (0.53, 0.61)* | 1.75 (1.69, 1.81)** |
| Access to media | | |
| No | Ref | Ref |
| Yes | 0.79 (0.77, 0.82)* | 1.60 (1.57, 1.63)** |
| Employment status | | |
| Non-working | Ref | Ref |
| Working | 0.90 (0.88, 0.92)* | 0.85 (0.84, 0.86)* |
| Parity | | |
| Nullipara | Ref | Ref |
| Primipara | 0.99 (0.94, 1.04) | 0.94 (0.90, 0.97) |
| Multipara | 0.92 (0.88, 0.96)* | 1.06 (1.02, 1.09)* |
| Contraceptive use | | |
| Not using | Ref | Ref |
| Using | 0.95 (0.92, 0.97)* | 1.22 (1.20, 1.25)** |
| Decision maker on women's health | | |
| Woman alone | Ref | Ref |
| Jointly with partner | 1.08 (1.04, 1.12)* | 0.94 (0.92, 0.96)* |
| Others[+] | 1.07 (1.03, 1.11)* | 0.87 (0.85, 0.89)* |
| Household wealth | | |
| Poor | Ref | Ref |
| Middle | 0.83 (0.80, 0.86)* | 1.24 (1.22, 1.27)** |
| Rich | 0.73 (0.71, 0.76)* | 1.49 (1.46, 1.52)** |
| Place of residence | | |
| Urban | 0.83 (0.80, 0.85)* | 1.66 (1.63, 1.69)* |
| Rural | Ref | Ref |
| Region | | |
| Africa | Ref | Ref |
| Asia | 1.48 (1.44, 1.53)* | 0.97 (0.95, 0.97)* |
| Others[++] | 0.43 (0.40, 0.47)* | 1.97 (1.91, 2.03)** |

[+]Others: husband only; someone else; other DHS category.

[++]Others: Latin America, Caribbean, and Europe.

*P-value < 0.05.

**P-value < 0.01. "Normal Weight" was the base category.

because women of households with middle and rich wealth indexes have relatively easier access to calorie-dense diets, which can lead to increased calorie intake. Additionally, their higher socioeconomic status tends to correlate with lower physical activity levels and increased sedentary behavior. Furthermore, psychosocial factors like stress associated with higher socioeconomic status can impact eating behaviors, further contributing to overweight and obesity [62].

In this analysis, women aged between 25–34 and 35–49 years were more likely to be overweight and obese than younger women. This aligns with the findings from the previous studies in different settings [63–66]. This might be due to biological changes and shifts in lifestyle as women age increases. Factors such as metabolic changes, reduced physical activity, and altered dietary habits might likely contributed to the observed result. Our finding also showed a lower risk of underweight, overweight and obesity among women who married at the age of 18 or older. The possible justification could be that delaying marriage allows women more time to develop healthier lifestyle habits, such as improved diet and regular exercise. Additionally, it correlates with higher education and socio-economic status, factors known to contribute to better health outcomes and reduced risk of weight-related issues [67].

Our study found a higher risk of overweight and obesity among women with primary, secondary, and higher education, alongside lower likelihood of underweight in these groups. These results align with previous studies across different settings [68–70]. The possible reason might be that higher educational attainment may contribute to better weight management practices by providing increased access to resources and knowledge about nutrition and healthy lifestyles. Furthermore, higher educational attainment often leads to greater economic stability, which can improve access to healthier food options. This improved access may contribute to a lower risk of underweight among women with higher education levels. However, on the other side, it can increase the risk of overweight and obesity due to potential changes in dietary habits and sedentary lifestyles (always in cars, with maid in houses, low levels of physical activity) associated with socio-economic status like [71].

Access to mass media was also identified as a significant factor linked with both undernutrition and overnutrition. The study found that media access was associated with a higher risk of obesity and overweight and a lower risk of underweight. Similarly, previous studies in both developed and developing nations [72–75] have reported comparable associations. Potential explanations include the promotion of unhealthy foods and sedentary behaviors via media, which can contribute to weight gain and obesity. Unrealistic body ideals portrayed in the media may also lead to body image issues and disordered eating, impacting both obesity and underweight trends. However, access to media also creates an opportunity for health literacy on dietary practices, potentially reducing the risks of malnutrition, and influencing cultural norms regarding dietary habits and weight management practices [76].

We found that employed women had a lower likelihood of undernutrition and overnutrition compared to their non-employed counterparts. Studies conducted in Malaysia, Ethiopia, and Mexico have reported similar findings [77–79]. The possible explanation could be that employed women experience higher stress levels and time constraints, leading to irregular eating habits and reliance on convenient but less nutritious food choices compared to their counterparts.

Similar to studies in both high- and low-income countries [23,80–82], our analysis showed that multiparous women have a higher risk of overweight and obesity than primipara and nullipara women. This could be due to the cumulative effect of weight gain across successive pregnancies, where weight gained during each pregnancy is not completely lost before the next pregnancy. Additionally, the demands of childcare and family responsibilities may reduce opportunities for physical activity and healthy meal planning, further contributing to the higher likelihood of overweight and obesity among multipara [80].

Contraceptive use was also found to be associated with overweight and obesity. The study showed that women using contraceptives are more likely to be overweight or obese compared to non-users. The current findings align with previous studies [83–85]. This can be explained by the fact that hormonal contraceptives alter hormone levels, potentially influencing metabolism, fat deposition, and appetite regulation, which promote weight gain. Additionally, reduced concern about pregnancy-related weight changes among contraceptive users may lead to decreased attention to diet and exercise, further contributing to a higher risk of overweight and obesity [86,87].

Furthermore, our analysis revealed that women whose healthcare relied solely on their husbands' decisions or who made joint decisions with their partners were more likely to be underweight compared to other women. Consistently, previous studies have shown similar associations [88–90]. This might be because societal norms and gender roles limit women's autonomy in practicing healthy behaviors and access to nutritious food and healthcare. These constraints, compounded by cultural expectations and lower social support within decision-making frameworks, ultimately contribute to inadequate nutrition and healthcare utilization, leading to poorer health outcomes such as underweight.

In addition, it was found that compared to African women, women in Latin America, the Caribbean, and Europe had a higher risk of being overweight and obese, while women in Asia had the lowest risk. Additionally, women in Asia were at higher risk of underweight than their African counterparts. Likewise, previous studies in LMICs have shown similar trends in overweight and obesity [91–93]. The possible reason could be that in rapidly urbanizing and economically growing regions like Latin America, the Caribbean, and Europe, increased sedentary behavior, higher consumption of processed foods, and reduced physical activity contribute to rising obesity rates. Conversely, in less developed settings like Asia and Africa, traditional diets and more active lifestyles may help maintain a lower risk of obesity [94].

Urban residence was also significantly associated with an increased risk of overweight and obesity. Women residing in urban areas exhibited a higher risk of overweight and obesity compared to their rural counterparts. This is consistent with the previous studies in SSA [58], Nigeria [95], Myanmar [96], and Ethiopia [97]. The possible explanation could be that in rural areas, women's active engagement in agriculture helps limit weight gain compared to urban women. Additionally, rural lifestyles generally involve less exposure to sedentary behavior, modern transportation, reduced physical activity, and consumption of high-fat, energy-dense foods typical of urban settings [1].

## Strengths and limitations

The strengths of this analysis lie in the use of recent population survey data from 40 countries, and the application of advanced statistical methods, i.e., multilevel multinomial modeling to account for the nested structure of the DHS data and the polytomous nature of the outcome variable. However, it is important to acknowledge certain limitations when interpreting the study's findings. These include potential over or underestimation of estimates due to a large number of missing observations (274,180 missing), and differences in the survey year, number of countries and sample sizes across the regions included in the analysis might affect the regional estimates. The use of BMI for assessing nutritional status does not differentiate between excess fat and bone mass and does not provide information about the distribution of body fat. Additionally, the study did not examine the effect of physical activity, dietary patterns, and household food security status on the outcome variable since DHS does not gather data on these variables.

## Conclusion

The pooled prevalence of obesity and overweight among WRA in LMICs was 26.8%, while underweight affected 15.6%. Our analysis also revealed that the levels of overweight and obesity were higher than the rate for underweight, with wide variations across the countries and survey periods. Moreover, the result showed that factors like household wealth, maternal education, age, place of residence, contraceptive use, media access, healthcare decisions, and employment status bidirectionally influenced both underweight and overweight and obesity. Efforts to improve household wealth, maternal literacy and healthcare, and employment opportunities for women can have a positive impact on reducing both underweight and overweight and obesity. Given the considerable variations between countries and survey time, nutrition programs should be tailored to the specific needs and cultural contexts of different regions within the study settings. Furthermore, the study's findings suggest the need for regular monitoring and evaluation of the existing nutritional interventions to assess their effectiveness.

## Supporting information

**S1. STROBE Statement—checklist of items that should be included in reports of observational studies.**
(DOCX)

## Acknowledgments

The authors thank ICF International for granting access to the dataset used in this study.

## Author contributions

**Conceptualization:** Kusse Urmale Mare, Beriso Furo Wengoro, Begetayinoral Kussia Lahole.

**Data curation:** Kusse Urmale Mare, Kebede Gemeda Sabo, Begetayinoral Kussia Lahole.

**Formal analysis:** Kusse Urmale Mare, Beriso Furo Wengoro, Begetayinoral Kussia Lahole.

**Investigation:** Kusse Urmale Mare.

**Methodology:** Kusse Urmale Mare, Kebede Gemeda Sabo, Beriso Furo Wengoro, Begetayinoral Kussia Lahole.

**Software:** Kusse Urmale Mare, Kebede Gemeda Sabo, Beriso Furo Wengoro, Begetayinoral Kussia Lahole.

**Validation:** Kusse Urmale Mare, Begetayinoral Kussia Lahole.

**Visualization:** Kusse Urmale Mare, Kebede Gemeda Sabo, Beriso Furo Wengoro, Begetayinoral Kussia Lahole.

**Writing – original draft:** Kusse Urmale Mare, Beriso Furo Wengoro, Begetayinoral Kussia Lahole.

**Writing – review & editing:** Kusse Urmale Mare, Kebede Gemeda Sabo, Beriso Furo Wengoro, Begetayinoral Kussia Lahole.

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
