## [Decision Letter · Decision Letter 0]

25 Sep 2024

PONE-D-24-26578Level of overweight and obesity surpassed underweight among women in 40 low and middle-income countries: findings from a multilevel multinomial analysis of population survey dataPLOS ONE

Dear Dr. Mare,

Thank you for submitting your manuscript to PLOS ONE. After careful consideration, we feel that it has merit but does not fully meet PLOS ONE’s publication criteria as it currently stands. Therefore, we invite you to submit a revised version of the manuscript that addresses the points raised during the review process.

The manuscript has been assessed by two reviewers and their comments are available below. They have requested more clarity in the study rationale and improvements to the reporting of the methodology and statistical analyses. Please review their comments and make the appropriate revisions. 

We look forward to receiving your revised manuscript.

Kind regards,

Emma Campbell, Ph.D

Staff Editor

PLOS ONE

Journal Requirements:

2. Please include captions for your Supporting Information files at the end of your manuscript, and update any in-text citations to match accordingly. Please see our Supporting Information guidelines for more information: http://journals.plos.org/plosone/s/supporting-information .

Reviewers' comments:

Reviewer's Responses to Questions

**Comments to the Author**

1. Is the manuscript technically sound, and do the data support the conclusions?

Reviewer #1: Yes

Reviewer #2: Yes

2. Has the statistical analysis been performed appropriately and rigorously? 

Reviewer #1: Yes

Reviewer #2: Yes

3. Have the authors made all data underlying the findings in their manuscript fully available?

Reviewer #1: Yes

Reviewer #2: Yes

4. Is the manuscript presented in an intelligible fashion and written in standard English?

Reviewer #1: Yes

Reviewer #2: Yes

5. Review Comments to the Author

Reviewer #1: This article provide advance information in the prevalence of overweight and obesity among women in 40 low- and middle-income countries exceeded underweight. This manuscript should goes the following suggested revision for improvement. Many sentences were started using gerund, authors are suggested to rewrite using simple or compound format. English should be checked by professionals

L17, L20, L23, Use the elaborate form of NCD, LMIC,, DHS

L28, IC to CI

L33, Add more core results such as women’s education, age, employment status, contraceptive use, region, place of residence, and media access on the risk of being underweight, overweight, and obese. If authors have no room, contract the conclusion section (L34-41) and add above mentioned results as much as possible.

L48, deposits to deposition.

L49-51, rewrite as, There are several linkages between obesity and health problems, including type 2 diabetes and heart disease, poor bone health, impaired reproductive function, and cancers of the reproductive organs [1,3,4] .

L77, World health organization (WHO)

L91, add sources (link/links or state the organization name of collector).

L168, end of the sentence (Table 1).

L174-175, L188-189, L210-211, delete. It is better not to include in the main text body. It has already been cited in the main text and given in the end of the manuscript which cause repetition and similar coments applies for figures captions in the main text body.

L187, Includes explanation of other parameters which was stated in Table 2.

L298, Add more concise form of result section.

L318, cite table of figure number according to your statement.

L723, Use different “*” or symbols to show different level of significance values. For instance, * for 0.0.5, ** for 0.01

Figure 3, vertical axis, Percent (%) to Percentage (%). Figure 3 has a similar trend in 2015 and 2022 for each parameters. Explain this phenomena in the result section (L220).

Reviewer #2: This is a well-written manuscript assessing abnormal nutritional indices, which describes the analysis of demographic and health survey data from 40 low- and middle-income countries utilizing a large sample size.

• Line number 86, please consider changing ‘reproductive age group women’ to ‘women of reproductive age’ (WRA) and replacing ‘women’ with ‘WRA’.

• Why are your study population women of reproductive age? What is the rationale?

• Citation is needed for the dependent variable measurement

• How was the variable ‘media exposure’ constructed?

• Indicate the outcome variable's reference category in Table 5. It looks like that "Normal Weight" was the base category.

• It is important to recognize the limitations of BMI in the strengths and limitations section, as it does not differentiate excess fat, muscle, or bone mass.

6. PLOS authors have the option to publish the peer review history of their article (what does this mean? ). If published, this will include your full peer review and any attached files.

**Do you want your identity to be public for this peer review?** For information about this choice, including consent withdrawal, please see our Privacy Policy .

Reviewer #1: No

Reviewer #2: No

---

## [Author Response · Author response to Decision Letter 0]

25 Nov 2024

Thank you for the opportunity to revise our manuscript. “Level of overweight and obesity surpassed underweight among women in 40 low and middle-income countries: findings from a multilevel multinomial analysis of population survey data (ID: PONE-D-24-26578)”. We have addressed the concerns raised by the reviewers using a point-by-point response as stated below. The amendments made to the manuscript have been presented using track change in the attachment titled “Revised manuscript with track changes”.

Response to Reviewer #1 Comments

Comment 1: This article provides advanced information on the prevalence of overweight and obesity among women in 40 low- and middle-income countries exceeded underweight. This manuscript should go the following suggested revision for improvement. Many sentences were started using gerund, authors are suggested to rewrite using simple or compound format. English should be checked by professionals

Response 1: Thank you for your suggestion. We have thoroughly revised the manuscript for grammar and editorial errors.

Comment 2: L17, L20, L23, Use the elaborate form of NCD, LMIC,, DHS

Response 2: Thank you very much for your suggestion. We have provided the elaborated form for the stated abbreviations in the revised manuscript.

Comment 3: L28, IC to CI

Response 3: Thank you and sorry for the editorial error. We have changed IC to CI.

Comment 4: L33, Add more core results such as women’s education, age, employment status, contraceptive use, region, place of residence, and media access on the risk of being underweight, overweight, and obese. If authors have no room, contract the conclusion section (L34-41) and add the above-mentioned results as much as possible.

Response 4: Thank you for your interesting comment. We have added the core results as per your suggestion in the abstract section of the revised manuscript.

Comment 5: L48, deposits to deposition

Response 5: Thank you very much. We have changed this in the revised manuscript.

Comment 6: L49-51, rewrite as, There are several linkages between obesity and health problems, including type 2 diabetes and heart disease, poor bone health, impaired reproductive function, and cancers of the reproductive organs [1,3,4].

Response 6: Thank you for your suggestion. We have incorporated the recommended change into the revised manuscript.

Comment 7: L77, World health organization (WHO)

Response 7: Thank you. We have considered the suggested change in the revision.

Comment 8: L91, add sources (link/links or state the organization name of collector).

Response 8: Thank you for your recommendation. We have provided a web link for the data source as suggested.

Comment 9: L168, end of the sentence (Table 1).

Response 9: Thank you. We have cited the table as per your recommendation.

Comment 10: L174-175, L188-189, L210-211, delete. It is better not to include in the main text body. It has already been cited in the main text and given in the end of the manuscript which cause repetition and similar comments applies for figures captions in the main text body.

Response 10: Thank you. We agree with your concern, but the journal guideline for preparing tables and figures suggests placing their captions in the main body of the manuscript as well as with each table and figure at the end of the main body.

Comment 11: L187, Includes an explanation of other parameters which was stated in Table 2.

Response 11: Thank you very much. We have added the explanation for other parameters as per your recommendation.

Comment 12: L298, Add a more concise form of the result section.

Response 12: Thank you. We have modified this section as suggested.

Comment 13: L318, cite table of figure number according to your statement.

Response 13: Thank you very much. We have cited the table and figure accordingly in the revision.

Comment 14: L723, Use different “*” or symbols to show different levels of significance values. For instance, * for 0.0.5, ** for 0.01

Response 14: Thank you. We have checked the regression table and provided a respective symbol to indicate different levels of significance as suggested in the revision.

Comment 15: Figure 3, vertical axis, Percent (%) to Percentage (%). Figure 3 has a similar trend in 2015 and 2022 for each parameter. Explain this phenomena in the result section (L220).

Response 15: Thank you for your suggestion. We have changed percent to percentage and explained the result observed between 2015 and 2022 in the revised manuscript.

Response to Reviewer #2 Comments

This is a well-written manuscript assessing abnormal nutritional indices, which describes the analysis of demographic and health survey data from 40 low- and middle-income countries utilizing a large sample size.

Comment 1: Line number 86, please consider changing ‘reproductive age group women’ to ‘women of reproductive age’ (WRA) and replacing ‘women’ with ‘WRA’.

Response 1: Thank you for your recommendation. We have made this change throughout the manuscript where it is appropriate.

Comment 2: Why are your study population women of reproductive age? What is the rationale?

Response 2: Thank you for your important concern. As we know, the Demographic and Health Survey (DHS) collects basic health and health-related information from women, men, and children in the sampled households. For the current analysis, we utilized the women's dataset, focusing on women of reproductive age (WRA), specifically those aged 15 to 49 years, within the households. Therefore, since we analyzed secondary data obtained from this source, the study population consisted of women of reproductive age.

Comment 3: Citation is needed for the dependent variable measurement

Response 3: Thank you very much. We have provided citation for the measurement of the dependent variable as suggested.

Comment 4: How was the variable ‘media exposure’ constructed?

Response 4: Thank you for your concern. Media exposure was generated using the frequency of watching television, listening to the radio, and reading newspapers, which have three response options (i.e. not at all, less than once a week, and at least once a week). Using the DHS definition, women who indicated that they watched television, listened to the radio, or read the newspaper at least once a week were classified as having media exposure, while those who did not meet this criterion were classified as not having media exposure. We have added this information under the “Variable and Measurement” section of the revised manuscript.

Comment 5: Indicate the outcome variable's reference category in Table 5. It looks like that "Normal Weight" was the base category.

Response 5: Thank you. We have indicated the base category of the outcome variable under Table 5 of the revision.

Comment 6: It is important to recognize the limitations of BMI in the strengths and limitations section, as it does not differentiate excess fat, muscle, or bone mass.

Response 6: Thank you for your important suggestion. We have added this limitation in the revised manuscript.

---

## [Decision Letter · Decision Letter 1]

13 Dec 2024

PONE-D-24-26578R1Level of overweight and obesity surpassed underweight among women in 40 low and middle-income countries: findings from a multilevel multinomial analysis of population survey dataPLOS ONE

Dear Dr. Mare,

Thank you for submitting your manuscript to PLOS ONE. After careful consideration, we feel that it has merit but does not fully meet PLOS ONE’s publication criteria as it currently stands. Therefore, we invite you to submit a revised version of the manuscript that addresses the points raised during the review process.

**See annotated manuscript for detailed comments for the manuscript:**1. The Methods Section casts doubts on the estimate: (i) It appears that authors are referring to many dependent variables yet there is only one (BMI, categorized); (ii) It is unclear in this section that overweight and obesity are treated as one category; (iii) while authors are doing multilevel modelling, it is unclear in this section what are Levl-1 and Level-2 variables. More importantly, it is unclear how Level-2 variables (also referred to as community-variables) they were constructed.2. It is unclear that analyses were performed on sub-samples of women of reproductive ages in all surveys sine DHSs don't collect weight and height for all women to compute BMI. As such, authors took the freedom to DROP cases with missing information on BMI. This is a "big mistake from sampling design point of view" because it will produce biased estimates. (see https://stats.oarc.ucla.edu/stata/faq/how-can-i-analyze-a-subpopulation-of-my-survey-data-in-stata/#:~:text=As%20stated%20in%20the%20Stata,are%20excluded%20from%20the%20analysis.)3. Finally, Discussion Section also needs refinement because it is redundant with Findings. The authors need to engage in discussion with previous studies on same topic while highlighting regional differences and implications for future health interventions.==============================

We look forward to receiving your revised manuscript.

Kind regards,

Zacharie Tsala Dimbuene, Ph.D.

Academic Editor

PLOS ONE

Reviewers' comments:

Reviewer's Responses to Questions

**Comments to the Author**

1. If the authors have adequately addressed your comments raised in a previous round of review and you feel that this manuscript is now acceptable for publication, you may indicate that here to bypass the “Comments to the Author” section, enter your conflict of interest statement in the “Confidential to Editor” section, and submit your "Accept" recommendation.

Reviewer #1: All comments have been addressed

Reviewer #2: All comments have been addressed

2. Is the manuscript technically sound, and do the data support the conclusions?

Reviewer #1: Yes

Reviewer #2: Yes

3. Has the statistical analysis been performed appropriately and rigorously? 

Reviewer #1: Yes

Reviewer #2: Yes

4. Have the authors made all data underlying the findings in their manuscript fully available?

Reviewer #1: Yes

Reviewer #2: Yes

5. Is the manuscript presented in an intelligible fashion and written in standard English?

Reviewer #1: Yes

Reviewer #2: Yes

6. Review Comments to the Author

Reviewer #1: The manuscript entitled "Level of overweight and obesity surpassed underweight among women in 40 low and middle-income countries: findings from a multilevel multinomial analysis of population

survey data". I appreciate authors efforts. The authors manage to response all the issues perfectly

Reviewer #2: The authors addressed all the comments and concerns raised in the first revision. I have no further comment.

7. PLOS authors have the option to publish the peer review history of their article (what does this mean? ). If published, this will include your full peer review and any attached files.

**Do you want your identity to be public for this peer review?** For information about this choice, including consent withdrawal, please see our Privacy Policy .

Reviewer #1: No

Reviewer #2: No

---

## [Author Response · Author response to Decision Letter 1]

23 Dec 2024

Thank you again for the opportunity to revise our manuscript “Level of overweight and obesity surpassed underweight among women in 40 low and middle-income countries: findings from a multilevel multinomial analysis of population survey data (ID: PONE-D-24-26578R1)“. We have addressed the concerns raised using a point-by-point response as stated below. The amendments made to the manuscript have been presented using track change in the attachment titled “Revised manuscript with track changes”.

Response to Editor’s comments

Comment 1: The Methods Section casts doubts on the estimate: (i) It appears that authors are referring to many dependent variables yet there is only one (BMI, categorized); (ii) It is unclear in this section that overweight and obesity are treated as one category; (iii) while authors are doing multilevel modeling, it is unclear in this section what are Levl-1 and Level-2 variables. More importantly, it is unclear how Level-2 variables (also referred to as community variables) were constructed.

Response 1: Thank you for your important concern and sorry for the editorial errors. We have amended the description of the dependent variable (BMI) and clarified that overweight and obesity were treated as one category. Additionally, we have specified level-1 and level-2 variables in the revised manuscript.

Comment 2: It is unclear that analyses were performed on sub-samples of women of reproductive ages in all surveys since DHSs don't collect weight and height for all women to compute BMI. As such, authors took the freedom to DROP cases with missing information on BMI. This is a "big mistake from the sampling design point of view" because it will produce biased estimates. (see https://stats.oarc.ucla.edu/stata/faq/how-can-i-analyze-a-subpopulation-of-my-survey-data-in stata/#:~:text=As%20stated%20in%20the%20Stata,are%20excluded%20from%20the%20analysis.)

Response 2: Thank you very much for pointing out this. We have stated that the analysis was performed on the sub-sample of women of reproductive age in all surveys. Regarding missing information, we agree with your concern that missing data can result in a biased estimate if not properly handled. For this reason, we have handled the missing data based on DHS guidelines (i.e. “women who were not weighed and measured and women whose values for weight and height were not recorded are excluded from both the denominator and the numerators.”) (see https://www.dhsprogram.com/pubs/pdf/DHSG1/Guide_to_DHS_Statistics_DHS-8.pdf on Page 11.10). Additionally, we have also acknowledged the drawback of missing data in the limitation section of the revision.

Comment 3: Finally, the Discussion Section also needs refinement because it is redundant with findings. The authors need to engage in discussion with previous studies on the same topic while highlighting regional differences and implications for future health interventions.

Response 3: Thank you for your important recommendation. We have removed redundant results throughout the discussion section as per your suggestion.

Response to comments in the annotated manuscript

Comment 1: Line 22-24: A methodological challenge arises when pooling data. I will look into the Methods Section but this needs to be clear on how Complex Survey Design was implemented in pooled data.

Response 1: Thank you for your concern. As we have stated in the data management and analysis sub-heading of the method section that we have used pooled/appended demographic and health survey data from 44 countries for the current analysis.

Comment 2: Line 26: abnormal nutritional indices (This means what?)

Response 2: Thank you. It was to indicate malnutrition (i.e. underweight and overweight and obesity). As a result, we have specified this concern in the revised manuscript.

Comment 3: Line 30-31: Now and again, this is the pitfall I usually see in pooled data since LMICs are not a homogeneous group on the one hand, and context matters on the other hand. Is this finding informative?

Response 3: Thank you very much for your concern. We have carried out a subgroup analysis to show regional differences in the revised manuscript. We also agree that pooling data from different countries has its drawbacks since the countries are not homogeneous and survey years are different. For this reason, we have considered these issues under the limitation section.

Comment 4: Line 33: women of middle and upper-class families?

Response 4: Thank you and sorry for the grammar error. We have changed this phrase into “women of families with middle and rich wealth indexes” accordingly in the revised manuscript.

Comment 5: Line 34: From primary to higher education? What about those with secondary education?

Response 5: Thank you for your question. For the analysis purpose, we have initially categorized educational status as no formal education, primary, secondary, and higher education. However, when we fit the regression model, the model omits the estimation of the relative risk ratio for the category “secondary education”. As a result, we recategorized this variable as “no formal education”, “primary education”, and “higher education (i.e. secondary + higher)”.

Comment 6: Line 37: You need to clarify the difference between middle- and upper-income families (By the way, income is not captured in DHS), and rich households.

Response 6: Thank you for your suggestion. In DHS, the household wealth index is categorized as poorest, poorer, middle, richer, and richest. For the analysis purpose, we have recategorized it as “poor”, “middle”, and “rich” based on the literature. Based on your suggestion, we changed these phrases to their correct DHS categories.

Comment 7: Line 43: modifiable risk factors of what?

Response 7: Thank you. It is just to indicate modifiable risk factors of both undernutrition and overnutrition. We have clarified this concern in the revised manuscript.

Comment 8: Line 62-64: You have furthermore and additionally in the same paragraph?

Response 8: Thank you very much. We have addressed this concern accordingly in the revised manuscript.

Comment 9: Line 72-74: You cannot list figures like this. Rewrite the sentence... using for instance: ranging from .... to ....

Response 9: Thank you for your interesting suggestion. We have revised this sentence as per your recommendation.

Comment 10: Line 77-80: same as above comment.

Response 10: Thank you for your interesting suggestion. We have revised this sentence as per your recommendation.

Comment 11: Line 103: Again, this poses a methodological concern that needs further scrutiny for reported findings to be trusted. 1) In the DHS, not all women in surveys were measured (weight, height) to compute BMI. As such, only a sub-sample of women has valid information on BMI. 2) How the complex survey design of DHS was treated in this case?

Response 11: Thank you for your important concern. Indeed, all women in the survey do not have BMI data (missing or not measured). For this reason, we have excluded cases with missing data based on DHS guidelines for handling missing data, and this issue was also acknowledged in the limitation section. Regarding the treatment complex survey design, we have used a multilevel model, which accounts for the hierarchical nature of the complex survey.

Comment 12: Line 107: This is not TRUE. The dependent variable is only one: BMI. Analytically, it was grouped into four categories: 1) underweight; 2) normal weight; 3) overweight; and 4) obese. Now, in many developing countries, obesity might represent a marginal percentage, which could lead to unstable statistical models. Please clarify your line of thought.

Response 12: Thank you very much for pointing out this important issue. As per your recommendation, we have clarified the dependent variable as BMI, with its respective categories (underweight, Normal, and overweight and obesity). We have also stated that overweight and obesity are treated as one category in the revised manuscript.

Comment 13: Line 112- Since you applied multilevel models, it helps readers if variables are grouped for each level

Response 13: Thank you for your suggestion. We have specified level-1 and level-2 variables in the revision.

Comment 14: Line 128-129: This is a serious methodological pitfall since it alters the survey design and produces "flawed" estimates

Response 14: Thank you very much for pointing out this. We have stated that the analysis was performed on the sub-sample of women of reproductive age from all surveys in the revision. Regarding missing information, we agree with your concern that missing data can result in a biased estimate if not properly handled. For this reason, we have handled the missing data based on DHS guidelines (i.e. “women who were not weighed and measured and women whose values for weight and height were not recorded are excluded from both the denominator and the numerators.”) (see https://www.dhsprogram.com/pubs/pdf/DHSG1/Guide_to_DHS_Statistics_DHS-8.pdf on Page 11.10). Additionally, we have also acknowledged the drawback of missing data in the limitation section of the revision.

Comment 15: Line 132: Level 1? Level 2? In short, how many LEVELS?

Response 15: Thank you very much. We have specified level-1 and level-2 variables under the “Independent variables” subsection of the revised manuscript.

Comment 16: Line 46: unknown as presented

Response 16: Thank you for your concern. We have replaced this with level-1 variables as per your suggestion in the “Independent variables” section.

Comment 17: Line 181: full term of WRA.

Response 17: Thank you. We have provided the extended form of WRA (i.e. women of reproductive age) in the revised result section.

Comment 18: Line 192: Reproductive and health-related characteristics. This appears for the first time.

Response 18: Thank you for your concern. For the descriptive purpose, we have separately presented the characteristics of the study participants as “Sociodemographic and reproductive-related characteristics”.

Comment 19: Line 211: In Figures 1-2, countries are thrown as they belonged to the same geographic region. Groupings need to be done.

Response 19: Thank you for your interesting comment. We have carried out subgroup analysis to show country-, regional, and overall estimates of underweight and overweight and obesity based on your suggestion. As a result, we have changed Figures 1 and 2 based on the reanalysis results.

Comment 20: Line 214: Finally, this contradicts what is said in the Methods Section regarding BMI (4 categories) but findings use only categories

Response 20: Thank you for your concern. We have modified the definition and categories of the outcome variable and stated that overweight and obesity are treated as one category as per our suggestion.

Comment 21: Line 266: As it stands now, the discussion is flawed: It is mostly a repetition of findings.

Response 21: Thank you for your important recommendation. We have removed redundant results throughout the discussion section as per your suggestion.

Comment 22: Line 267: It should be clear throughout the manuscript that overweight and obesity are treated as one category.

Response 22: Thank you for your suggestion. We have clarified that overweight and obesity were treated as one category throughout the revised manuscript.

Comment 23: Line 292: what does middle class mean? You use interchangeably income, class, households???? WHY, explain!

Response 23: Thank you very much and sorry for the editorial error. We have consistently used the appropriate terms for the household wealth index throughout the revised manuscript.

Comment 24: Line 313: what about women with secondary education

Response 24: Thank you for your question. For the analysis purpose, we have initially categorized educational status as no formal education, primary, secondary, and higher education. However, when we fit the regression model, the model omits the estimation of the relative risk ratio for the category “secondary education”. As a result, we re-categorized this variable as “no formal education”, “primary education”, and “higher education (i.e. secondary + higher)”.

Comment 25: Line 318-319: I expect that higher education leads to wealth, but at the same time leads to sedentary lifestyles (always in cars, with maids in houses, low levels of physical activity).

Response 25: Thank you for your interesting comment. We have added this explanation to the revised manuscript.

Comment 26: Line 346: Please be specific

Response 26: Thank you. We have amended this concern throughout the revised manuscript.

Response to Reviewers' comments

Reviewer #1: The manuscript entitled "Level of overweight and obesity surpassed underweight among women in 40 low and middle-income countries: findings from a multilevel multinomial analysis of population survey data". I appreciate the authors' efforts. The authors manage to respond to all the issues perfectly.

Response: Thank you for your appreciation and for accepting our manuscript in its current form.

Reviewer #2: The authors addressed all the comments and concerns raised in the first revision. I have no further comment.

Response: Thank you for accepting our manuscript in its current form.

---

## [Editor Report · Decision Letter 2]

8 Jan 2025

PONE-D-24-26578R2Level of overweight and obesity surpassed underweight among women in 40 low and middle-income countries: findings from a multilevel multinomial analysis of population survey dataPLOS ONE

Dear Dr. Mare,

Thank you for submitting your manuscript to PLOS ONE. After careful consideration, we feel that it has merit but does not fully meet PLOS ONE’s publication criteria as it currently stands. Therefore, we invite you to submit a revised version of the manuscript that addresses the points raised during the review process.

1) The presentation of independent variables is still unsatisfactory, and dependent variable as well. If overweight and obesity are treated as. one category, this should be done in ALL Tables through the manuscript. 2) The structure of the variables (Level-1) should follow the (i) women-; and (ii) household- variables in this order3) The justification of recording women's education in two categories (no education and higher) is less convincing. I suspect there is a coding inconsistency across countries leading to the error the authors are getting. 

We look forward to receiving your revised manuscript.

Kind regards,

Zacharie Tsala Dimbuene, Ph.D.

Academic Editor

PLOS ONE
---

## [Author Response · Author response to Decision Letter 2]

9 Jan 2025

Thank you again for the opportunity to revise our manuscript “Level of overweight and obesity surpassed underweight among women in 40 low and middle-income countries: findings from a multilevel multinomial analysis of population survey data (ID: PONE-D-24-26578R2)”. We have addressed the concerns raised using a point-by-point response as stated below. The amendments made to the manuscript have been presented using track change in the attachment titled “Revised manuscript with track changes”.

Response to Editor’s Comments

Comment 1: The presentation of independent variables is still unsatisfactory, and dependent variable as well. If overweight and obesity are treated as one category, this should be done in ALL Tables through the manuscript.

Response 1: Thank you very much. We have corrected the presentations of independent and dependent variables as per your suggestions throughout all tables in the revision.

Comment 2: The structure of the variables (Level-1) should follow the (i) women-; and (ii) household- variables in this order

Response 2: Thank you for your recommendation. We have presented the level-1 variables as per the suggested order in the revised tables.

Comment 3: The justification of recording women's education in two categories (no education and higher) is less convincing. I suspect there is a coding inconsistency across countries leading to the error the authors are getting.

Response 3: Thank you very much for your interesting recommendation and sorry for providing a misleading justification in the previous response. We have checked the coding of educational status for all countries as suggested. Based on this, we did reanalysis and revised the results and discussion sections accordingly. We appreciate you for this important critique.

Journal Requirements:

Comment: Please review your reference list to ensure that it is complete and correct. If you have cited papers that have been retracted, please include the rationale for doing so in the manuscript text, or remove these references and replace them with relevant current references. Any changes to the reference list should be mentioned in the rebuttal letter that accompanies your revised manuscript. If you need to cite a retracted article, indicate the article’s retracted status in the References list and also include a citation and full reference for the retraction notice.

Response: Thank you. We have reviewed the reference list and confirm that it is complete and correct and no retracted article has been included in the list.

---

## [Editor Report · Decision Letter 3]

13 Feb 2025

Level of overweight and obesity surpassed underweight among women in 40 low and middle-income countries: findings from a multilevel multinomial analysis of population survey data

PONE-D-24-26578R3

Dear Dr. *Kusse Urmale Mare* ,

We’re pleased to inform you that your manuscript has been judged scientifically suitable for publication and will be formally accepted for publication once it meets all outstanding technical requirements.

Kind regards,

Zacharie Tsala Dimbuene, Ph.D.

Academic Editor

PLOS ONE
---

## [Editor Report · Acceptance letter]

PONE-D-24-26578R3

PLOS ONE

Dear Dr. Mare,

I'm pleased to inform you that your manuscript has been deemed suitable for publication in PLOS ONE. Congratulations! Your manuscript is now being handed over to our production team.

Kind regards,

on behalf of

Prof. Zacharie Tsala Dimbuene

Academic Editor

PLOS ONE